# Modelling the impact of the tier system on SARS-CoV-2 transmission in the UK between the first and second national lockdowns

Daniel J Laydon ![ORCID],[1] Swapnil Mishra ![ORCID],[1] Wes R Hinsley,[1] Pantelis Samartsidis,[2] Seth Flaxman,[3] Axel Gandy,[4] Neil M Ferguson,[1] Samir Bhatt[1]

[1]Department of Infectious Disease Epidemiology, MRC Centre for Global Infectious Disease Analysis, Jameel Institute for Disease and Emergency Analytics, Imperial College London, London, UK
[2]MRC Biostatistics Unit, Cambridge Institute of Public Health, University of Cambridge, Cambridge, UK
[3]Department of Mathematics and Data Science Institute, Imperial College London, London, UK
[4]Department of Mathematics, Imperial College London, London, UK

**Correspondence to**
Dr Daniel J Laydon;
d.laydon@imperial.ac.uk

## ABSTRACT

**Objective** To measure the effects of the tier system on the COVID-19 pandemic in the UK between the first and second national lockdowns, before the emergence of the B.1.1.7 variant of concern.

**Design** This is a modelling study combining estimates of real-time reproduction number $R_t$ (derived from UK case, death and serological survey data) with publicly available data on regional non-pharmaceutical interventions. We fit a Bayesian hierarchical model with latent factors using these quantities to account for broader national trends in addition to subnational effects from tiers.

**Setting** The UK at lower tier local authority (LTLA) level. 310 LTLAs were included in the analysis.

**Primary and secondary outcome measures** Reduction in real-time reproduction number $R_t$.

**Results** Nationally, transmission increased between July and late September, regional differences notwithstanding. Immediately prior to the introduction of the tier system, $R_t$ averaged 1.3 (0.9–1.6) across LTLAs, but declined to an average of 1.1 (0.86–1.42) 2 weeks later. Decline in transmission was not solely attributable to tiers. Tier 1 had negligible effects. Tiers 2 and 3, respectively, reduced transmission by 6% (5%–7%) and 23% (21%–25%). 288 LTLAs (93%) would have begun to suppress their epidemics if every LTLA had gone into tier 3 by the second national lockdown, whereas only 90 (29%) did so in reality.

**Conclusions** The relatively small effect sizes found in this analysis demonstrate that interventions at least as stringent as tier 3 are required to suppress transmission, especially considering more transmissible variants, at least until effective vaccination is widespread or much greater population immunity has amassed.

## INTRODUCTION

There is substantial evidence supporting the effectiveness of government-mandated social distancing (so-called 'lockdown') measures[1–5] in suppressing SARS-CoV-2 transmission. The massive economic and social cost of these measures in the UK motivated the less socially disruptive tier system, introduced on 14 October 2020.[6] Its aim was to provide a consistent set of COVID-19 control measures

### Strengths and limitations of this study

► This is the first study to measure the effects of the UK tier system for SARS-CoV-2 control at national and regional level.
► The model makes minimal assumptions and is primarily data-driven.
► There is insufficient statistical power to estimate the effects of individual interventions that comprise tiers, or their interaction.

with geographical flexibility. Tiers consist of multiple non-pharmaceutical interventions (NPIs) and were determined by lower tier local authorities (LTLAs) in response to their local transmission intensity. Tiers were inconsistently defined across LTLAs; for example, some LTLAs left gyms and fitness centres open under tier 1, while most did not. There is currently little evidence on the effectiveness or otherwise of the UK tier system, which ran until the second national lockdown on 5 November 2020, before the emergence of the more transmissible variant of concern B.1.1.7 and before tier 4 was enacted.

Estimating the effect of the tiers on the underlying transmission is challenging due to lags between their implementation and any change in cases, deaths and results of serological surveys. Therefore, estimating effect sizes by using only raw data can produce spurious results. Semimechanistic modelling that combines a transmission model with statistical modelling of transmission provides an alternative. There are two common approaches, estimating effect sizes directly within a semimechanistic transmission model[1] or estimating effect sizes in two stages,[4] by first non-parametrically estimating $R_t$ and then performing a more classic regression analysis. Both approaches have their strengths and weaknesses from a statistical

viewpoint. From a computational perspective, however, full Bayesian modelling of a semimechanistic model across more than 300 LTLAs is not tractable. In this work, we therefore adopt a two-stage approach. Leveraging our existing LTLA transmission intensity estimation platform,[7] we first estimate $R_t$ (the rate of transmission; ie, the number of secondary infections per infection at time $t$) using data on cases, deaths and serology. We model $R_t$ as a weekly random process, making no assumptions on the effect or timing of interventions, and incorporate a wide range of data synthesis mechanisms. We further collate publicly available data on NPIs and tiers at LTLA level. We combine these two sources of information to fit a Bayesian hierarchical model and estimate the effects of the UK tier system between the first and second national lockdowns.

## METHODS

### Collation of intervention data by LTLA

We compiled a list of the interventions implemented by each LTLA over time between 1 July 2020 and 5 November 2020. Data were obtained by monitoring the following government sources:

► https://www.gov.uk/government/collections/local-restrictions-areas-with-an-outbreak-of-coronavirus-covid-19.
► https://www.gov.uk/guidance/full-list-of-local-covid-alert-levels-by-area.

The interventions and tiers that LTLAs were subject to were also amended accordingly.

### Backdating tier classification

Since the tier classification came into place on 14 October 2020, a strict evaluation of the effects of tiers would limit our analysis to only 21 days until the second national lockdown on 5 November. However, the interventions that comprise a given tier were in place earlier. We therefore retrospectively assign a tier rating to LTLAs from 1 July 2020, when restrictions of the first national lockdown were lifted for the majority of the UK. The challenge is that the exact interventions that were implemented under tiers 1, 2 and 3 varied first between LTLAs, and second over time. We determine the interventions that comprise a given tier (and thus which tier a given LTLA would have been in before the creation of the tier system) using the number of LTLAs that implemented a given intervention on the day tiers 1–3 were applied. Our study period ends before the second national lockdown and the implementation of tier 4.

Figure 1 shows the number of LTLAs that implemented a given intervention when nominally under tiers 1, 2 and 3, at the earliest date they were introduced. Four interventions are implemented in all 310 LTLAs when under tier 1: (1) limiting indoor gatherings to at most six people; (2) limiting outdoor gatherings to at most six people; (3) curfew of 22:00 for hospitality venues; and (4) instruction to work from home where possible. A further three interventions ('travel discouraged', 'no indoor mixing', 'overnight stays discouraged') are considered to be part of tier 1 for 30 LTLAs. However, as this is minority of LTLAs, and as these interventions are always present under tier 2, we include them in the definition of tier 2. Tier 3 is the most consistently defined set of interventions and adds the following two interventions to tier 2: 'residents cannot leave the local area' and 'pubs and bars closed table service only'. Closure of gyms is included in tier 3 for six LTLAs, but this is insufficiently frequent to warrant inclusion in the backdating of tier 3.

### Real-time reproduction number estimates

Full details of the $R_t$ estimation can be found in Mishra *et al.*[7] Briefly, however, the method estimates transmission by calculating backwards from observed deaths (day of death), cases and serological survey data while simultaneously allowing for the time lag between infection and death. Infection fatality ratios and infection ascertainment

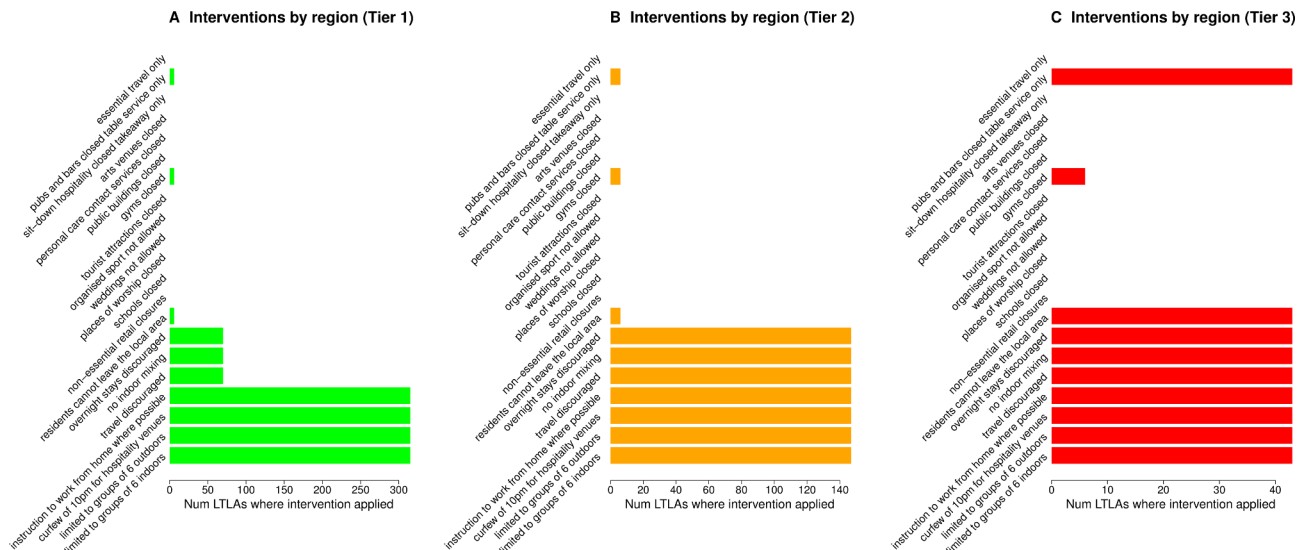

**Figure 1** Number of lower tier local authorities (LTLAs) applying interventions at introduction of tier 1 (A), tier 2 (B) or tier 3 (C).

rates are also calibrated nationally. The method accounts for noise in the data, noise in the stochastic process, lags in reporting and day of the week variation.

## Hierarchical model

With $R_t$ estimates for weekly time points $t=1, ..., \tau$ ($\tau$=19 weeks between 1 July 2020 and 4 November 2020) and for each individual LTLA indexed by $m = 1, ... M$ ($M = 310$ individual local authorities), we use a regression model to estimate the effect size of the introduction of tiers. Specifically, for each LTLA $m$, we have a response variable $y_m \in R^T$ of $R_t$ values, and a binary design matrix $x_m \in 1R^{\tau \times 3}$, where each column corresponds to the backdated tier 1, 2 or 3, and the indicator function is a binary toggle for whether an LTLA is in a given tier on a given date. Defining tiers cumulatively (eg, if an LTLA is in tier 2 then it is also in tier 1) makes little difference to our results. We propose a latent factor model that estimates the effect of tiers, but that also accounts for confounding from other factors. We name these other factors secular trends, and they represent changes in transmission due to individual or wider societal behavioural changes along with LTLA idiosyncrasies and demographics, which are not part of the formal tier system.

The linear model specification is:

$$y_m \sim Normal(\mu_m, \sigma)$$
$$\mu_m = \exp(\lambda \alpha_m^T - \beta x_m^T)$$
$$\beta \sim Normal^+\left(0, \phi_1\right)$$
$$\alpha_m \sim Normal^+\left(0, \phi_2\right)$$
$$\lambda \sim Normal^+\left(0, \phi_3\right)$$
$$\phi_1, \phi_2, \phi_3 \sim Normal^+\left(0, 2\right)$$
$$\sigma \sim Normal^+\left(0, 2\right)$$

In the absence of LTLA-mandated interventions, a given LTLA $m$ has a trend $\lambda \alpha_m^T$ in its reproduction number over time $R_t$. The above model is multiplicative: tiers reduce the $R_t$ trend $\lambda \alpha_m^T$ by a factor of $\exp(-\beta x_m^T)$, for $\beta \in R^3$. The secular trends $\lambda \alpha_m^T$ can be factorised into two basis functions with internal dimension $b$, that is, a national trend $\lambda \in R^{\tau \times b}$ and a regional scaling $\alpha_m \in R^b$. This factorisation enforces shared trends across all LTLAs. The internal dimension $b$ can be interpreted as the number of factors or categories that each LTLA has (to varying extents), in addition to the tiers, that may influence transmission intensity. $b$ can also be interpreted as the number of ways in which regional $R_t$ values can differ from national values. Model flexibility and complexity increase with $b$, and we set $b=2$. Note that the national trend $\lambda$ has the same values for all LTLAs, while each $\alpha_m$ basis differs by LTLA $m$. The first column in the basis function $\lambda$ is an intercept, and the second are $\tau$ independent parameters. Normal shrinkage priors are used for $\lambda$ to balance overfitting and underfitting. Conceptually, the intercept component of the basis matrix $\lambda$ accounts for different LTLA starting values in $R_t$, which could be due to different demographics, behaviour or contact patterns.

The second basis $\alpha_m$ scales and moves the national secular trend $\lambda$ for a given LTLA $m$. The secular trend broadly corresponds to changes over time due to behaviour and other national policies that are independent from the tiers (such as 'the rule of six'). However, the absolute impact of the national trend can vary by LTLA. That is, the overall shape of the trend is shared nationally, but its magnitude and starting value can differ across LTLAs. As well as being relatively parsimonious, this formulation captures the interdependence of $R_t$ values between LTLAs, while not assuming any *a priori* relationship between LTLAs.

It is possible to allow for a larger internal dimension value $b$, but given that our $R_t$ estimates and backdated tiers consist of only 22 weeks of data for each LTLA, increasing $b$ risks overfitting. With greater internal dimension, the model is too flexible and so the data will struggle to separate the effects of the tiers $\beta x_m^T$ from the underlying trend in each LTLA's transmission $\lambda \alpha_m^T$. A similar problem arises if the priors for the $\beta$ parameters are not restricted to non-negative values. Such additional model complexity spuriously nullifies the effects of tiers. Alternative bases for $\lambda$ such as b-splines could be used, but we opted for a non-structured basis to limit *a priori* assumptions. Partial pooling of $\beta$, whereby the effects of tiers would differ by LTLA, was investigated but did not produce significant differences between LTLAs.

We conducted a sensitivity analysis on the priors of the hyperparameters $\varphi_1$, $\varphi_2$, $\varphi_3$ and $\sigma$ of the hierarchical model (which, respectively, inform the values for tier coefficients, the LTLA-specific scaling of the secular trend, the secular trend itself and the random noise around the resulting $R_t$ values). In our main model these are given as positive normal distributions with SD of 2. However, allowing the SD to take values of 0.5, 1 or 2.5 made negligible difference to our results. Further, we investigated heavy tailed Cauchy distributions. While this resulted in a posterior distribution with greater curvature, it made negligible difference to estimated tier coefficients. Our results are therefore robust to changes in the prior, and we therefore opted for normal distributions with SD of 2 for our main model, as this prior is wide and uninformative.

We fit to $R_t$ estimates by LTLA over time, and not to cases and deaths. NPIs and tiers ultimately affect transmission, which in turn affects cases and deaths only after a delay. Therefore, regressing on the time-varying reproduction number is preferable as such delays do not need to be incorporated into the model.

Fitting was performed using the Bayesian software platform Stan,[8] via the R package 'rstan'[9] and R V.3.6.3,[10] using 2000 iterations with a warm-up of 500 iterations across 10 chains. R-hat and other diagnostics indicated a fully converged model. All model code and data sets are available from either the authors, or alternatively at https://github.com/ImperialCollegeLondon/covid19model.

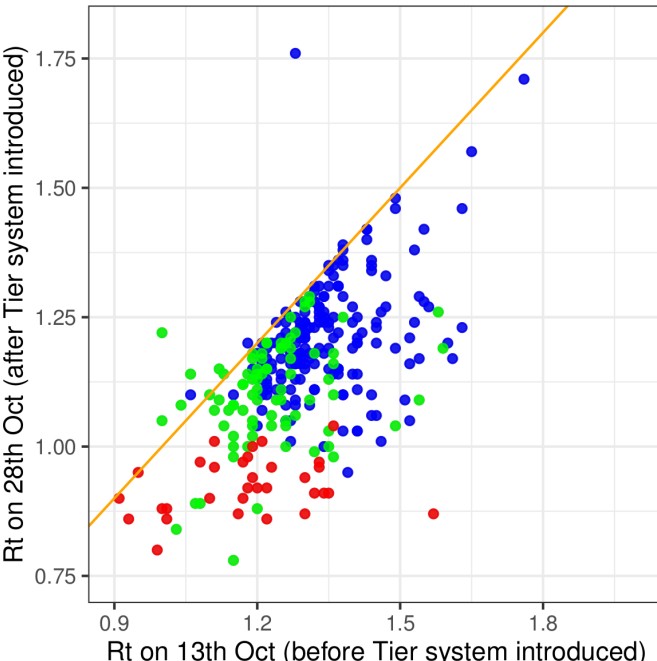

**Figure 2** Real-time reproduction number $R_t$ by lower tier local authority 1 day before and 2 weeks after introduction of the tier system. Blue points are tier 1, green points are tier 2 and red points are tier 3. Line of equality is shown in orange.

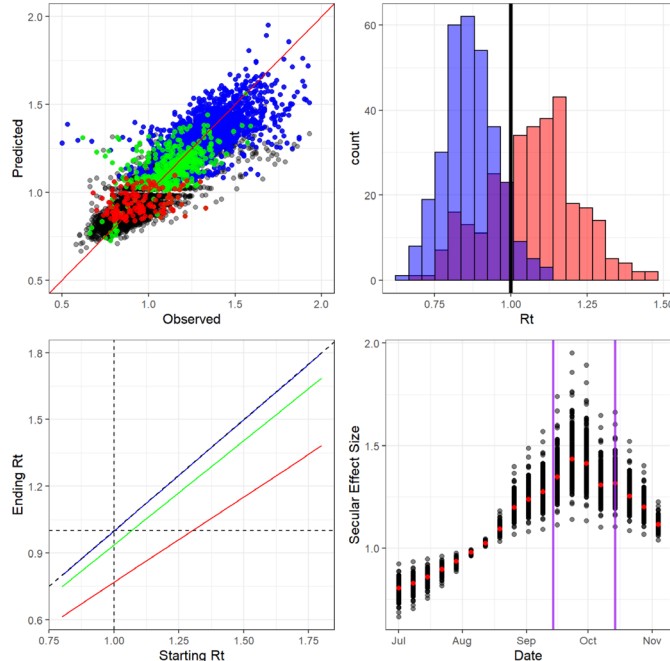

**Figure 3** (Top left) Observed versus predicted real-time reproduction number $R_t$. Mean absolute error of 10% correlation of 63%. Blue points are tier 1, green points are tier 2 and red points are tier 3. Black points are dates before the tier system introduction. (Top right) Overlapping histograms of $R_t$ on 4 November 2020 for all LTLAs. Red bars are actual $R_t$ values, blue bars are $R_t$ if all LTLAs entered tier 3. (Bottom left) Black line is $R_t$=1, the implied starting and ending $R_t$ for a given tier. Blue line is tier 1, green line is tier 2 and red line is tier 3. Dotted lines are the 1 to 1 line and $R_t$=1. (Bottom right) Secular national trend from the latent factor models. Each dot for each week represents the mean effect size for a given LTLA and the red dots are the mean across all LTLAs. The first purple line corresponds to the rule of six introduced and the second purple line corresponds to the formal introduction of the tier system. LTLA, lower tier local authority

## RESULTS

Figure 2 shows the $R_t$ estimates the day before the tier system was introduced and 2 weeks afterwards. On 13 October, $R_t$ averaged 1.3 (0.9–1.6) across LTLAs, but by 28 October had declined to 1.1 (0.86–1.42).

Between 1 July and 5 November 2020, $R_t$ took an average value of 1.13 (95% percentile between 0.74 and 1.62) across all LTLAs. The latent factor secular trend accounts for much of the variation in $R_t$. The mean absolute error for our model is 8.7%, and without latent factors is 14%. The overall model fit (figure 3, top left) suggests our model can reproduce trends in $R_t$ relatively accurately (87% correlation). The secular trends (figure 3, bottom right) suggest that transmission increased through to mid-September, when it began to reach a plateau. Transmission started to decline from late September, that is, before the tier system was introduced on 14 October. This implies that reduction in transmission is partly attributable to secular trends, independent of tiers. Tier coefficients can be interpreted as a relative reduction in $R_t$. Therefore if $R_t$=1, in the absence of tiers, the reproduction number after entering a given tier is given by the exponentiated coefficients in table 1. That is, for any value of $R$, the reduction from tier $i$, relative to the secular trend, is simply $R \times \exp(-\beta_i)$ for tier coefficient $\beta_i$ (table 1). We detected no discernible effect from tier 1 in addition to the secular trend, with $\beta_1$=0.001 ($1.3 \times 10^{-5}$–0.002), scaling $R_t$ by 1. Tier 2 is slightly more effective and scales $R_t$ by a factor of 0.94 (0.93–0.95). However, the effects of tier 3 are significant, as it scales $R_t$ by 0.77 (0.75–0.79).

It is important to stress that these estimates do not imply that the interventions comprising tier 1 (eg, working from home recommendations or limiting indoor gatherings to six people or less) would have zero effect, but rather that they have no additional effect beyond the secular trend. For example, tier 1's prohibition of indoor or outdoor gatherings larger than six people was introduced on 14 October, after the more general 'rule of six' that was already in place from 14 September.

We investigated what the effects on transmission would have been if all LTLAs had entered tier 3. On 4 November 2020, only 28% of LTLAs were managing to suppress their epidemic with $R_t$ <1, with 95% of LTLAs having $R_t$ values ranging from 0.79 to 1.35. Figure 3 (top right) shows the overall effect on $R_t$ if all LTLAs had entered tier 3 on 4 November 2020. If all LTLAs had entered tier 3 on 5 November, we estimate that 288 LTLAs (93%) would have reduced $R_t$ to less than 1 (95% credible interval of $R_t$ values ranges between 0.66 and 1.03). In fact, only 90 LTLAs (29%) managed to do so. These results suggest tier 3 would have had a substantial effect on transmission, but

**Table 1** Coefficient reductions from a given tier

| Variable | Mean | 2.5% Credible Interval | 97.5% Credible Interval |
|---|---|---|---|
| Tier 1 | 1 | 1 | 1 |
| Tier 2 | 0.94 | 0.93 | 0.95 |
| Tier 3 | 0.77 | 0.75 | 0.79 |

Note these have been exponentiated and can be interpreted by directly multiplying them with a given $R_t$.

$R_t$, real-time reproduction number.

would not conclusively reduce $R_t$ below 1 in every LTLA. It should also be noted that if all LTLAs had been in tier 3 on a different date, the effect would not have been the same due to secular trend changes.

## DISCUSSION
### Principal findings
We estimated the effects of tiers 1, 2 and 3 between the first and second national lockdowns in the UK. We estimated of the local time-varying reproduction number $R_t$ using our established semimechanistic model. We combined these estimates with data detailing the timing of NPIs and tiers at LTLA level to use as inputs in a Bayesian hierarchical model with a latent factor analysis.

Our approach aimed to account for broader secular trends at the national level in addition to the tiers at LTLA level. The model reproduced $R_t$ estimates reasonably well and broadly captured their variation over time and variation between LTLAs. The national secular trend increased between early July and late September, notwithstanding regional differences and control measures applied. The national trend began to plateau from mid-September (before the introduction of the tier system) and declined shortly after, implying that transmission reduction was not solely attributable to tiers. We find that tier 1 has almost no effect on transmission beyond the secular national trend, and that tier 2 yielded only minor reductions in transmission. However, tier 3 was more effective, reducing transmission by an average of 23%. We estimated that, had tier 3 been in effect throughout all LTLAs, the real-time reproduction number $R_t$ would have been reduced to below 1 in 93% of LTLAs (288 LTLAs) at the start of the second national lockdown, as opposed to the 29% of LTLAs (90 LTLAs) that actually did manage to reduce $R_t$ to below 1 at this time.

### Limitations
Our approach has a number of limitations. While the latent factors account for confounding to some degree, the coefficients we estimate are non-causal and therefore provide only associative effects. We do not consider the interaction of interventions, but merely their joint effect as mandated through the tier system. In generating counterfactuals our model makes a 'difference in differences'

counterfactual assumption, which has previously been shown to have limitations[11] arising from the assumption of parallel trajectories. It is important to note that the effect sizes we model quantify the instantaneous and constant impact of the tiers on $R_t$, whereas the effects of tiers may vary over time, perhaps with a lag before they take effect or with a waning of efficacy.

Our backdating of tiers is imperfect: government announcement of a given tier may have an additional effect beyond that of the particular NPIs within that tier. However, it remains a reasonable approximation that enables the analysis of NPIs in finer detail than lockdown. It would be useful to measure the effects of the specific interventions that and comprise the tiers, as this would enable more targeted measures for COVID-19 control. However, the data are unfortunately insufficiently powered to make such inferences.

Further, our priors assume that tiers will either reduce transmission or be ineffective, but do not allow tiers to increase $R_t$. However, given that reduced human contact must reduce transmission, this is a reasonable assumption. We also do not propagate uncertainty in the reproduction number in the hierarchical model, as the high number of LTLAs renders this computationally intractable. The development of frameworks that can propagate uncertainty reliably is an important priority for future work.

## CONCLUSIONS
Our analysis focused on the period between the first and second national lockdowns, before the emergence of the more transmissible B.1.1.7 variant and before the roll-out of vaccination. This focus avoids potential confounding factors of vaccination—it is impossible that we have falsely attributed the effects of vaccination to the impact of tiers. While mass vaccination will enable less stringent interventions over time, the emergence of more transmissible variants will render tiers 1 and 2 even less likely to be able to control SARS-CoV-2 transmission.

NPIs will remain necessary to control SARS-CoV-2 transmission, particularly in light of newer more transmissible variants, and at least until an efficacious and effective vaccine becomes widely available or much greater population immunity has amassed. The relatively small effect sizes found in this analysis caution against expecting dramatic additional reductions in $R_t$ from interventions less extreme than tier 3.

**Acknowledgements** We thank Joni Kirk and Pantelis Hadjipantelis of the Joint Biosecurity Centre, Department of Health and Social Care, and Shaun Seaman of the MRC Biostatistics Unit, Cambridge Institute of Public Health for helpful discussions.

**Contributors** Conceptualisation: DJL, SM, NMF, SB. Data curation: DJL, SM, WRH, SF, AG, NMF, SB. Formal analysis: DJL, SM, SF, AG, NMF, SB. Funding acquisition: NMF, SB. Investigation: DJL, SM, NMF, SB. Methodology: DJL, SM, SF, PS, AG, NMF, SB. Project administration: NMF, SB. Resources: DJL, SM, WRH, SF, AG, NMF, SB. Software: DJL, SM, WRH, NMF, SB. Supervision: NMF, SB. Validation: DJL, SM, SF,

AG, NMF, SB. Visualisation: DJL, NMF, SB. Writing-original draft: DJL, SB. Writing-review and editing, and review of final draft: all authors.

**Funding** DJL and NMF acknowledge funding from Vaccine Efficacy Evaluation for Priority Emerging Diseases (VEEPED) grant (ref NIHR: PR-OD-1017–20002) from the National Institute for Health Research. SB acknowledges the NIHR BRC Imperial College NHS Trust Infection and COVID-19 themes, and the Academy of Medical Sciences Springboard award (ref SBF004\1080). DJL, SM, WRH, NMF and SB are supported by Centre funding (grant MR/R015600/1) from the UK Medical Research Council under a concordat with the UK Department for International Development, the NIHR Health Protection Research Unit in Modelling Methodology and Community Jameel.

**Disclaimer** The funders of the study had no role in study design, data collection, data analysis, data interpretation or writing of the report. The corresponding author had full access to all the data in the study and had final responsibility for the decision to submit for publication.

**Competing interests** None declared.

**Patient and public involvement** Patients and/or the public were not involved in the design, or conduct, or reporting, or dissemination plans of this research.

**Patient consent for publication** Not required.

**Provenance and peer review** Not commissioned; externally peer reviewed.

**Data availability statement** Data are available in a public, open access repository. Data are available upon reasonable request. All model code and data sets are available from either the authors, or alternatively at https://github.com/ImperialCollegeLondon/covid19model.

**ORCID iDs**
Daniel J Laydon http://orcid.org/0000-0003-4270-3321
Swapnil Mishra http://orcid.org/0000-0002-8759-5902

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
