## [Reviewer comments · BMJ Open]

This paper was submitted to a another journal from BMJ but declined for publication following peer review. The authors addressed the reviewers' comments and submitted the revised paper to BMJ Open. The paper was subsequently accepted for publication at BMJ Open.

ARTICLE DETAILS

TITLE (PROVISIONAL)	Modelling the impact of the Tier system on SARS-CoV-2 transmission in the UK between the first and second national lockdowns
AUTHORS	Laydon, Daniel; Mishra, Swapnil; Hinsley, Wes; Samartsidis, Pantelis; Flaxman, Seth; Gandy, Axel; Ferguson, Neil; Bhatt, Samir

VERSION 1 - REVIEW

REVIEWER	Kinloch, Emma Salivary Gland Cancer UK
REVIEW RETURNED	24-Dec-2020

GENERAL COMMENTS	This is a very important and timely paper. Whilst I am unable to comment specifically on the methods, the results and conclusions are relevant and important to patients and carers. The authors acknowledge that there is no analysis of individual components of the interventions, nor the interactions between these components. Could there be a recommendation to conduct this analysis if possible? If there is to be suggestion of greater use of 'more stringent' lock-down measures i.e. 'at least tier 3' as suggested in the paper - it might also be helpful to strengthen the points around the vaccination programme being rolled out e.g. increased understanding / use of the value of the first vaccine over using both doses which could considerably reduce the time taken for the population to be 'safe'. This is hugely important when considering the mental impact and economic impact of these 'more stringent' measures.
--

REVIEWER	Wu, Joseph University of Hong Kong, Infectious Disease Epidemiology Group, School of Public Health, Li Ka Shing Faculty of Medicine
REVIEW RETURNED	21-Jan-2021

GENERAL COMMENTS	In this study, the author estimated the effect of the Tier system on reducing the reproductive number of COVID-19
---

	in the UK between the first and second national lockdowns. My major comments are as follows:  1. Overall the methodology is sensible. However, the validity of the results strongly depends on whether the inferred secular trends, which account for the majority of the variation in the observed R_t, are valid. Can the authors provide external evidence to substantiate the validity of the inferred secular trends? 2. It is not clear whether the R_t estimates here have adjusted for the delay between infection and reporting (Gostic et al, PLOS Comp Bio). 3. The uncertainty of the estimates in Table 1 is surprisingly low. How sensitive are the regression results to the values of the hyperparameters in the priors? 4. The LTLAs are being treated as independent patches in the regression model. One would expect the R_ts from LTLAs with relative strong human mobility connections to have some degree of dependence. The authors should elaborate on how robust their results are against such dependency.
--	---

VERSION 1 – AUTHOR RESPONSE

Reviewer 1:

This is a very important and timely paper.

Whilst I am unable to comment specifically on the methods, the results and conclusions are relevant and important to patients and carers.

The authors acknowledge that there is no analysis of individual components of the interventions, nor the interactions between these components. Could there be a recommendation to conduct this analysis if possible?

We agree that quantifying the effects of the individual interventions that comprise a given Tier (and their interactions) would be especially helpful in guiding COVID-19 control policy. We did investigate models that included a term for every intervention listed in Figure 2. However, we found that models would not converge, and the resulting uncertainty of individual intervention effect sizes was too large to be useful, even after back-dating Tiers. We conclude that, unfortunately, the data is insufficiently powered to infer the effects of individual interventions. These points are discussed on page 16, lines 302-305.

If there is to be suggestion of greater use of 'more stringent' lock-down measures i.e. 'at least Tier 3' as suggested in the paper - it might also be helpful to strengthen the points around the vaccination programme being rolled out e.g. increased understanding / use of the value of the first vaccine over using both doses which could considerably reduce the time taken for the population to be 'safe'. This is hugely important when considering the mental impact and economic impact of these 'more stringent' measures.

We of course agree that as vaccination is rolled out, the need for NPIs will (thankfully) lessen. We have focussed on the period between the first and second lockdowns, as this is before vaccination roll-out, when vaccination could not possibly bias our estimates (i.e. falsely attributing effects of vaccination to effects of

Tiers). Our analysis is therefore useful in evaluating UK government policy, and will additionally be helpful in considering new variants that evade natural or vaccine-induced immunity, or indeed future pandemics.

We have included greater consideration of this point in the discussion (page 16, lines 307-313).

Reviewer 2:

In this study, the author estimated the effect of the Tier system on reducing the reproductive number of COVID-19 in the UK between the first and second national lockdowns. My major comments are as follows:

1. Overall the methodology is sensible. However, the validity of the results strongly depends on whether the inferred secular trends, which account for the majority of the variation in the observed R_t , are valid. Can the authors provide external evidence to substantiate the validity of the inferred secular trends?

We agree that external validation of the effects of Tiers and the secular trends we estimate could potentially be useful. However, this is not possible in practice due to data limitations. Incidence of deaths (or cases) and serology, and therefore the real-time reproduction number, can be viewed as the aggregate of interventions/Tiers and underlying transmission. It is not possible to attribute which factors drive trends in R_t without an exhaustive assessment of all factors, such as behaviour, media, environment etc. A common way to deal with this limitation is to use latent processes, and this is what we use. Disaggregating the effects of interventions/Tiers from underlying transmission is the main aim of our analysis, which to our knowledge is the first to do so at the level of Tiers, as opposed to national lockdown.

We further cannot validate our secular trend using daily deaths or cases. The reproduction numbers to which we fit our hierarchical model are themselves derived from daily deaths or case and death data. So, using these data to validate our estimates would use circular logic. We can however say that the hierarchical model reproduces estimates of R_t well (page 11, lines 218-219).

Nonetheless, the reviewer's point is important. The estimates of the effects of Tiers are influenced by the secular trend, as well as the extent to which each LTLA differs from the secular trend. We have included a sensitivity analysis (as suggested in another comment), which supports the robustness of our estimates. This analysis can be found in the methods on page 8, lines 167-177.

2. It is not clear whether the R_t estimates here have adjusted for the delay between infection and reporting (Gostic et al, PLOS Comp Bio).

We thank the reviewer for drawing our attention to this oversight. The R_t estimates are based on daily case and death data, and serology data. They include lags in reporting as well as estimates of infection ascertainment. We have added a short section in the methods (page 5, lines 106-112) to clarify these points and to give greater detail on the R_t estimates. We also cite the work that describes the method in full, however please let us know if any further detail would be helpful in this manuscript.

3. The uncertainty of the estimates in Table 1 is surprisingly low. How sensitive are the regression results to the values of the hyperparameters in the priors?

We have added a section detailing a sensitivity analysis on the hyperparameters of the hierarchical model (page 8, lines 167-177). Varying firstly the standard deviation of the priors on the hyperparameters, and secondly the prior distributions themselves, we find that our estimates are not sensitive to the choice of prior.

We were also initially surprised at the narrow posterior estimates of Tier effects, and this is what prompted us to use a wide prior in order to reflect this uncertainty *a priori*. However, the data drives the posterior contraction in this coefficient uncertainty and leads to narrow intervals. We have checked and this posterior distribution is well calibrated and reflects the uncertainty correctly. The addition of the above sensitivity analysis provides assurance that our estimates are robust, and that our default model can therefore use a wide and essentially uninformative prior of $Normal^*(0,2)$.

We would prefer not to “cram” the Results section with multiple, highly similar Tables, but would be happy to include these as supplementary tables if the reviewers or editors would prefer.

We thank the reviewer for raising this point. We believe the manuscript is stronger as a result.

4. The LTLAs are being treated as independent patches in the regression model. One would expect the R_t s from LTLAs with relative strong human mobility connections to have some degree of dependence. The authors should elaborate on how robust their results are against such dependency

We agree that LTLAs are not independent of one another. However, in our model, we do not treat LTLAs as strictly independent. Rather, transmission in each LTLA is influenced by the secular trend. LTLAs can differ in their magnitude and starting values of R_t , but the overall form of the R_t values is shared amongst LTLAs. The exception to this is of course the Tiers, which do differ between LTLAs, and these properties are explicitly preserved in our model.

There are many ways in which LTLAs are interdependent. Our relatively parsimonious and data driven approach does not assume that differences or dependencies between LTLAs in their underlying transmission are due to any one factor, such as mobility. We have given greater consideration to this point (page 7, lines 148-155).

Because our approach gave reasonable model fits, and because estimates were in line with SPI-M consensus estimates, we would prefer not to add additional model complexity where in our view it is not needed. We are of course happy to reconsider if the reviewer disagrees.

VERSION 2 – REVIEW

REVIEWER	Hatzakis, Angelos University of Athens, Athens, Greece, National Retrovirus Reference Center, Department of Hygiene, Epidemiology and Medical Statistics
REVIEW RETURNED	09-Mar-2021

GENERAL COMMENTS	It is a very interesting study providing methods to assess the effectiveness of Tier system. Each country uses different Tier system and comprehensive methods to analyze them is going to offer a lot in Tier' s evaluation. The methods described are probably more interesting than the findings and I wish the authors could
--

	provide a less technical description of their methods in a general medical journal.
--	---

REVIEWER	Merler, Stefano Bruno Kessler Foundation
REVIEW RETURNED	16-Mar-2021

GENERAL COMMENTS	Laydon and colleagues presented an interesting manuscript on the impact of the tier system on SARS-CoV-2 transmission in the UK. The work has undergone several revisions that seem to have been adequately addressed by the authors. Unfortunately, the study design is not ideal to assess the impact of tiers on transmission (due to tiers being enacted to mitigate transmission when needed, rather than to assess their impact). However, authors acknowledged such limitation, stating that “[a more complex model of the secular trend] may spuriously nullify the effect of tiers” and that “estimate[s] are noncausal”. Furthermore, results are consistent with findings in other countries (e.g. Manica et al., medRxiv) in showing that more aggressive tiers were able to increasingly reduce the reproduction number. Minor comments:  • In the model equations, it is not clear to me why the beta parameters are also sampled from a positive normal distribution. If a tier has zero effect, the estimate will have to be distributed between positive and negative values, while it is currently forced to have a positive effect, however small; • Figure 2 has five colors (red, lighter red, olive, yellow and green), but only three are specified in the legend and capture. I guess it may be due to the superposition of colors via transparencies, but it is really difficult to read and I think it should be amended; maybe it would be better to use side-by-side barplots rather than stacked/overlapping ones. Although I understand that part of the tiers have been imputed by the authors, I think the figure and the paragraph commenting it are more appropriate in the Methods section rather than in the Results; • Considering that LTLAs seem to be quite small geographical entities, and that Rts were estimated from deaths and reported cases, I expect a large variability in the mean estimates for smaller LTLAs, especially for periods of lower incidence. Did the authors check whether excluding the most uncertain data points (e.g. those for which $sd/mean > a$ given threshold) may help obtaining more robust estimates? • Lines 257-258: I would not equate $R_t < 1$ with “suppressing the epidemics”; • Line 259: typo in LTLAS (rather than LTLAs)
--

VERSION 2 – AUTHOR RESPONSE

Reviewer: 1

Prof. Angelos Hatzakis, University of Athens, Athens, Greece

Comments to the Author:

It is a very interesting study providing methods to assess the effectiveness of Tier system. Each country uses different Tier system and comprehensive methods to analyze them is going to offer a lot in Tier's evaluation. The methods described are probably more interesting than the findings and I wish the authors could provide a less technical description of their methods in a general medical journal.

We thank the reviewer for their positive comments. We have added additional descriptive text in the methods (e.g. lines 150-151).

Reviewer: 2

Dr. Stefano Merler, Bruno Kessler Foundation

Comments to the Author:

Laydon and colleagues presented an interesting manuscript on the impact of the tier system on SARS-CoV-2 transmission in the UK. The work has undergone several revisions that seem to have been adequately addressed by the authors. Unfortunately, the study design is not ideal to assess the impact of tiers on transmission (due to tiers being enacted to mitigate transmission when needed, rather than to assess their impact). However, authors acknowledged such limitation, stating that “[a more complex model of the secular trend] may spuriously nullify the effect of tiers” and that “estimate[s] are noncausal”. Furthermore, results are consistent with findings in other countries (e.g. Manica et al., medRxiv) in showing that more aggressive tiers were able to increasingly reduce the reproduction number.

Minor comments:

- In the model equations, it is not clear to me why the beta parameters are also sampled from a positive normal distribution. If a tier has zero effect, the estimate will have to be distributed between positive and negative values, while it is currently forced to have a positive effect, however small.*

We investigated non-constrained priors for the Tier effects, but found these had identifiability issues with the latent factor component. Our prior puts most of the mass of tier effects at small values but excludes the possibility that the tier system could have increased transmission. Given that COVID-19 transmission must be reduced with as human contact is reduced, we think this is a reasonable assumption. We have included this point in the Methods (lines 177-178) and Discussion (lines 292-294).

- *Figure 2 has five colors (red, lighter red, olive, yellow and green), but only three are specified in the legend and capture. I guess it may be due to the superposition of colors via transparencies, but it is really difficult to read and I think it should be amended; maybe it would be better to use side-by-side barplots rather than stacked/overlapping ones. Although I understand that part of the tiers have been imputed by the authors, I think the figure and the paragraph commenting it are more appropriate in the Methods section rather than in the Results.*

We have moved paragraph in question to the Methods, and have renumbered Figures accordingly. We have amended Figure 2 (now Figure 1) to use three barplots side-by-side.

- *Considering that LTLAs seem to be quite small geographical entities, and that R_t s were estimated from deaths and reported cases, I expect a large variability in the mean estimates for smaller LTLAs, especially for periods of lower incidence. Did the authors check whether excluding the most uncertain data points (e.g. those for which $sd/mean > a$ given threshold) may help obtaining more robust estimates?*

This is an important point and we agree with the reviewer. In an ideal scenario we could jointly estimate the Tier effects within a full transmission model. This is currently intractable in most reliable probabilistic programming frameworks due to the high number of LTLAs. Propagating the uncertainty in R_t is important for future work, but due to the study period we consider, we do not believe our model is especially susceptible to high stochasticity in R_t . We have added this point to the Limitations section of the Discussion.

- *Lines 257-258: I would not equate $R_t < 1$ with “suppressing the epidemics”;*

We agree and have amended the text accordingly (now lines 267-268).

- *Line 259: typo in LTLAS (rather than LTLAs)*

Fixed now. Thank you.